# Delta Decompression for MoE-based LLMs Compression

**Hao Gu** [* 1]  **Wei Li** [* 2]  **Lujun Li** [* 1]  **Qiyuan Zhu** [1]  **Mark Lee** [2]  **Shengjie Sun** [3]  **Wei Xue** [1]  **Yike Guo** [1]

## Abstract

Mixture-of-Experts (MoE) architectures in large language models (LLMs) achieve exceptional performance, but face prohibitive storage and memory requirements. To address these challenges, we present $D^2$-MoE, a new delta decompression compressor for reducing the parameters of MoE LLMs. Based on observations of expert diversity, we decompose their weights into a shared base weight and unique delta weights. Specifically, our method first merges each expert's weight into the base weight using the Fisher information matrix to capture shared components. Then, we compress delta weights through Singular Value Decomposition (SVD) by exploiting their low-rank properties. Finally, we introduce a semi-dynamical structured pruning strategy for the base weights, combining static and dynamic redundancy analysis to achieve further parameter reduction while maintaining input adaptivity. In this way, our $D^2$-MoE successfully compacts MoE LLMs to high compression ratios without additional training. Extensive experiments highlight the superiority of our approach, with over 13% performance gains than other compressors on Mixtral|Phi-3.5|DeepSeek|Qwen2 MoE LLMs at 40~60% compression rates. Codes are available in https://github.com/lliai/D2MoE.

## 1. Introduction

Recent advances in Large Language Models (LLMs) increasingly favor Mixture of Experts (MoE) (Cai et al., 2024) architectures for their ability to scale model capacity through specialized expert networks while maintaining computational efficiency via sparse activation. The success of MoE is evident in recent LLMs like DeepSeek-V3 (DeepSeek-AI

*Table 1.* Comparison of our method with other MoE compressors. Diversity means retaining individual per-expert information. Maximum ratio denotes the maximum parameter compression ratio.

| Method | Strategy | Structured | Train-free | Diversity | Max-Ratio |
|---|---|---|---|---|---|
| NAEE (2024b) | Prune | ✓ | ✗ | ✗ | 50% |
| MoE-Compress (2024) | Prune | ✗ | ✓ | ✗ | 50% |
| MoE-Pruner (2024) | Prune | ✗ | ✗ | ✗ | 50% |
| MoE-$I^2$ (2024) | Prune | ✓ | ✗ | ✗ | 55% |
| MC-SMoE (2023b) | Merge | ✗ | ✗ | ✗ | 75% |
| HC-SMoE (2024) | Merge | ✓ | ✓ | ✗ | 50% |
| EEP (2024a) | Merge | ✓ | ✓ | ✗ | 75% |
| $D^2$-**MoE (Ours)** | Delta | ✓ | ✓ | ✓ | 80% |

et al., 2024) and MiniMax-01 (MiniMax et al., 2025), which demonstrate unprecedented capabilities in language understanding and generation tasks. Despite their compelling advantages, MoE LLMs face critical challenges in practical deployment scenarios (Tang et al., 2024; Zhong et al., 2024; Hwang et al., 2024). **Their substantial parameter footprint, coupled with considerable memory overhead** from storing multiple expert weights (Song et al., 2023), creates significant barriers to resource-constrained environments.

To address these challenges, MoE compression methods have recently gained significant attention. As illustrated in Table 1, current approaches broadly categorized into expert pruning and expert merging methods. **(1) Expert pruning approaches,** represented by MoE-Pruner (Xie et al., 2024), NAEE (Lu et al., 2024a), and MoE-I²(Yang et al., 2024), implement inter-expert pruning and intra-expert weight sparsification. While these approaches achieve significant parameter reduction, they often result in substantial performance degradation due to the irreversible loss of expert knowledge. The direct removal of expert weights compromises the model's specialized capabilities, frequently necessitating additional fine-tuning to partially recover performance. **(2) Expert merging methods**, on the other hand, aim to consolidate multiple experts into fewer, more compact representations. Methods like EEP (Liu et al., 2024a), MC-SMoE (Li et al., 2023b), and HC-SMoE (Chen et al., 2024) develop various weighting schemes for weighted summation of different experts' weights. While these approaches preserve more information than direct pruning, it introduces new challenges. The merging process *assumes significant overlap in expert functionalities, but in practice, experts often possess distinct, complementary specializations.* This leads

---

*Equal contribution [1]Hong Kong University of Science and Technology [2]University of Birmingham [3]AISpeech Co., Ltd.. Correspondence to: Mark Lee <M.G.Lee@bham.ac.uk>, Yike Guo <yikeguo@ust.hk>.

*Proceedings of the 42$^{nd}$ International Conference on Machine Learning*, Vancouver, Canada. PMLR 267, 2025. Copyright 2025 by the author(s).

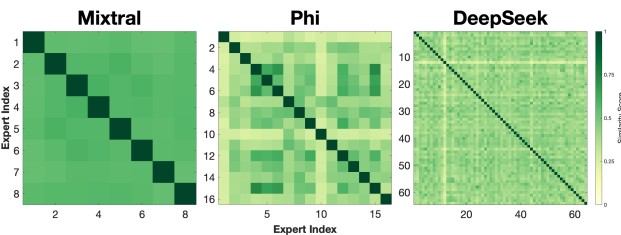

*Figure 1.* Centered Kernel Alignment (CKA) similarity of experts weights of Mixtral-8x7B, Phi-3.5-MoE, DeepSeekMoE-16B-Base.

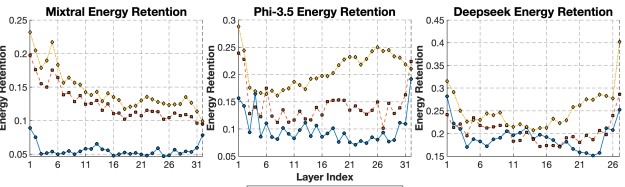

*Figure 2.* Single values energy retention of experts original weights, merged base weights and delta weights (difference in original weights and merged base weights) from Mixtral-8x7B, Phi-3.5-MoE, DeepSeekMoE-16B-Base.

to a fundamental dilemma: experts with similar weights can be effectively merged, but those with dissimilar yet important weights resist efficient compression, resulting in either suboptimal compression ratios or performance degradation. These challenges present the question: **How can we design new frameworks beyond pruning and merging methods in effectively balancing compression and preserving expert diversity?**

*"Diversity is not about how we differ. Diversity is about embracing one another's uniqueness."*

*— Ola Joseph*

As the quote goes, recent fine-tuning methods (Ping et al., 2024) quantize delta weights between fine-tuned and original models to effectively capture both similarities and variations. Inspired by these successes, we investigate whether it is possible to recycle the difference (delta) weights that are always discarded during expert merging to maintain performance without introducing excessive computational or memory overhead. Specifically, we brainstorm the idea to efficiently reallocate these abandoned delta weights (differences between merged expert weights and original weights) to preserve the diversity and specialization of experts. To explore this, we conduct two key experiments to analyze the properties of expert weights in MoE LLMs: **(1)** We evaluate expert similarity using centered kernel alignment (CKA) metrics. As shown in Figure 1, the similarity between different expert weights consistently falls within the 0.3 to 0.5 range. This indicates a moderate overlap in their feature spaces, suggesting that *while some aspects of their weights can be merged, preserving expert diversity remains crucial.* **(2)** We examine distributions of single values energy retention for different expert weight decompositions (detailed in Appendix B.2). As illustrated in Figure 2, *the larger singular values of energy retentions in the delta weights show that most of the matrix's information is concentrated in a small number of singular vectors, indicating a strong low-rank structure.* This shows that these delta weights can be efficiently approximated using low-rank decomposition methods without excessive degradation of information. These findings underscore that reutilizing delta weights to expert merging is a promising way for MoE compression

that balances efficiency, diversity, and performance.

Building on these insights, we develop $D^2$-MoE, a novel compression framework to address the growing challenges of parameter redundancy, memory overhead, and storage inefficiency in MoE LLMs while preserving model performance and scalability. Rather than directly removing or merging experts, our approach strategically decomposes expert weights into a shared base weight, which captures the commonalities across all experts, and a delta weight, which encodes the expert-specific variations. This decomposition not only reduces redundancy but also facilitates efficient compression of delta weights by exploiting their inherent low-rank structure. To ensure that the shared base weight accurately represents the most critical information across experts, $D^2$-MoE incorporates a Fisher-weighted averaging mechanism. This approach computes the shared base weight by weighting each expert's contribution based on its Fisher importance, which quantifies the sensitivity of the model's parameters to the input data. By prioritizing the contributions of the most important experts, Fisher-weighted averaging balances the trade-off between redundancy reduction and representational fidelity. To further compress the delta weights, $D^2$-MoE employs a truncation-aware SVD method that integrates activation statistics into the decomposition process. This method adjusts the singular value truncation threshold based on the input activation patterns, ensuring that essential information is preserved while compressing delta weights. Finally, $D^2$-MoE proposes semi-dynamical structured pruning on the shared base weight, combining static and dynamic pruning phases to eliminate redundant parameters while adapting to the input distribution in real-time. With these new schemes, our $D^2$-MoE enjoys the benefits of being structured and acceleratable, requiring no extra training, preserving expert diversity and performance, and realizing high compression ratios (see Table 1).

Our extensive experimental evaluation highlights the exceptional performance of $D^2$-MoE across multiple state-of-the-art MoE language models and a wide range of benchmarks. For models like Mixtral-8×7B and DeepSeekMoE-16B-Base, $D^2$-MoE achieves the lowest perplexity on language modeling datasets and the highest average accuracy

on reasoning benchmarks, even at high compression ratios (*e.g.* 0.52 average accuracy at 60% compression for Mixtral-8×7B, compared to 0.36 for NAEE). On large-scale models such as Phi-3.5-MoE and Qwen2-57B-A14B, $D^2$-MoE maintains strong performance, delivering accuracy close to the original model while significantly outperforming methods like MoE-$I^2$. The consistent superiority of $D^2$-MoE across diverse MoE LLMs and tasks demonstrates its general applicability and effectiveness in preserving expert specialization and task performance while achieving substantial efficiency gains, setting a new standard for MoE compression.

## 2. Related Work

**Mixture of Experts Compression** methods (see Table 1) reduce parameter redundancy and minimize storage in MoE models. For example, MoE-Pruner (Xie et al., 2024) achieves compression by pruning weights based on their activations and router importance. However, these unstructured methods typically provide only limited inference acceleration. For structured pruning, NAEE (Lu et al., 2024a) skips non-redundant experts and trims unimportant weight connections, while MoE-I$^2$(Yang et al., 2024) combines inter-expert pruning with intra-expert low-rank decomposition. Yet, these methods involve a serious loss of expert knowledge, requiring additional fine-tuning. Our approach differs from these methods by avoiding the direct removal of experts and no need for retraining. Expert merging methods like EEP (Liu et al., 2024a) introduce a two-stage pipeline where experts are first pruned and then merged into consolidated representations. Similarly, MC-SMoE (Li et al., 2023b) groups experts based on routing policies and merges each group into a single expert. However, merging experts inherently reduces the diversity of the model, potentially harming its ability to generalize across diverse input distributions. Methods like HC-SMoE (Chen et al., 2024) mitigate retraining requirements but are still limited by the trade-off between compression and preserving the model's capacity. In contrast, our framework strategically isolates shared knowledge into a base weight while retaining expert-specific variations as delta weights. In addition, our semi-dynamic pruning and other techniques also do not exist in the previous methods.

**Delta compression** in LLMs has emerged as a critical technique to reduce the storage and computational costs of deploying multiple fine-tuned models by compressing the differences (delta weights) between a base model and its fine-tuned variants. Recent advancements, GPT-Zip (Isik et al., 2023) and BitDelta Liu et al. (2024b) successfully quantize the delta weights into ultra-low bit. Delta-CoMe (Ping et al., 2024) employs mixed-precision quantization to the varying singular vectors of decomposed delta weights. An-

other approach, DeltaZip (Yao & Klimovic, 2023) develops a multi-tenant serving system by compressing delta weights. In contrast to these quantization and system-level works, **we not only first introduce delta compression into MoE compression, but also propose new techniques like MoE-specific SVD and semi-dynamic pruning,** achieving the optimal performance-efficiency trade-off.

## 3. Methodology

### 3.1. Delta Compression in MoE LLMs

**MoE Formulation.** MoE architectures enhance the capacity and efficiency of LLMs by employing expert-based Feed-Forward Network (FFN) layers. The output $y$ of the MoE-FFN layer for an input $x$ is computed as:

$$y = \sum_{i=1}^{N} G(x)_i \cdot E_i(x), \tag{1}$$

where $N$ is the total number of experts, $G(x) \in \mathbb{R}^N$ represents the gating weights, and $E_i(x)$ is the output of the $i$-th expert. The sparsity is achieved through a top-$k$ selection mechanism:

$$G(x) := \text{Softmax}(\text{TopK}[x \cdot W_g]) \tag{2}$$

where $\text{TopK}[\cdot]$ selects the $k$ experts with highest gating weights, and Softmax normalizes their weights. This results in a sparse activation of experts for efficiency. Each expert $E_i$ is a standard FFN layer, typically consisting of two or three fully connected layers. These experts constitute the majority of the weights in MoE models (*e.g.* 96% for Mixtral-8x7B), making them the most focus of compressors (*e.g.*, MC-MoE and our $D^2$-MoE).

**Experts Delta Decomposition.** MoE models are highly parameterized due to the presence of multiple experts, leading to significant redundancy among expert weights. Directly compressing these weights often results in performance degradation, as shared structures across experts are not fully exploited. To address this, we introduce a **delta compression** strategy that decomposes the weights of each expert into two components: a shared base weight that captures commonalities across all experts and a delta weight that encodes expert-specific variations. This decomposition reduces redundancy, facilitates efficient compression, and minimizes performance loss. Let $W_i \in \mathbb{R}^{m \times n}$ represent the weight matrix of the $i$-th expert, where $m$ and $n$ denote the input and output dimensions of the FFN layer, respectively. We express $W_i$ as the sum of a shared base weight $W_b$ and an expert-specific delta weight $\Delta W_i$:

$$W_i = W_b + \Delta W_i. \tag{3}$$

Here, $W_b \in \mathbb{R}^{m \times n}$ is shared across all experts, and $\Delta W_i \in \mathbb{R}^{m \times n}$ represents the unique characteristics of the

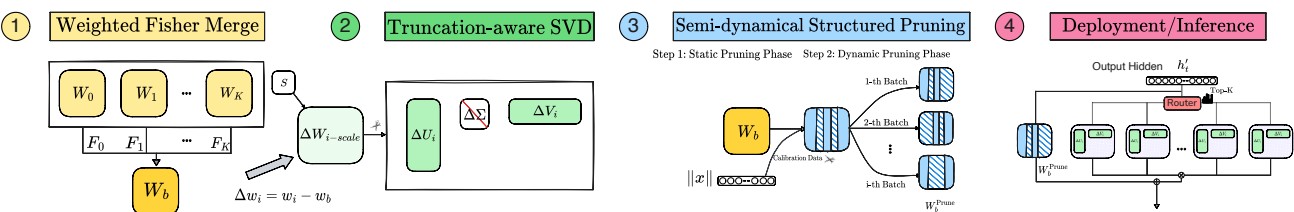

*Figure 3.* Overall Process of $D^2$-MoE. We first merge original expert weights into a shared base weight, weighted according to their Fisher importance. Then, we derive delta weights and compress them using Singular Value Decomposition (SVD). Finally, we apply a two-step pruning strategy: static column-wise pruning followed by dynamic column-wise pruning to further optimize the base weight.

$i$-th expert. By separating the shared and expert-specific components, we ensure that $W_b$ captures the common structure, reducing redundancy in the delta weights $\Delta W_i$.

### 3.2. Experts Fisher Merging

To effectively derive base weights that represent the shared knowledge across subset of experts $\mathcal{K} \subseteq \{1, \ldots, N\}$ (of size $K = |\mathcal{K}|$) while retaining essential diversity, our goal is to compute a merged base weight $W_b$ that minimizes redundancy and preserves the critical information required for downstream tasks. Traditional methods, such as simple averaging, compute the merged weight as the element-wise arithmetic mean of the weights of all experts, performed as $W_b = \frac{1}{K} \sum_{i \in \mathcal{K}} W_i$. Although simple averaging is computationally efficient, it fails to consider the varying importance of different experts. This can lead to under-representation of critical weights in the base weight $W_b$ and increase the difficulty of compressing delta weights $\Delta W_i$ in later stages. To address this, we incorporate the Fisher information matrix (Matena & Raffel, 2022), which captures the importance of the parameters of each expert in the merging process. Our merging function uses Fisher-weighted averaging to compute the base weight $W_b$. The importance of each expert is quantified using the Fisher information matrix, which measures the sensitivity of the model's parameters to the log-likelihood of the data. Specifically, the Fisher information for the $i$-th expert is given by:

$$F_i = \mathbb{E}_{x \sim D_i} \mathbb{E}_{y \sim p_\theta(y|x)} \left[ \nabla_{\theta_i} \log p_\theta(y|x)^2 \right], \quad (4)$$

where $D_i$ represents the data distribution handled by expert $i$, $p_\theta(y|x)$ is the predicted probability of label $y$ given input $x$, and $\nabla_{\theta_i} \log p_\theta(y|x)$ is the gradient of the log-likelihood with respect to the parameters $\theta_i$ of the $i$-th expert. Intuitively, $F_i$ measures the average magnitude of the gradient norm, with higher values indicating that the expert's parameters are more critical for the model's performance. Using the Fisher importance $F_i$, we compute the Fisher-weighted base weight $W_b$ as:

$$W_b = \frac{\sum_{i \in \mathcal{K}} F_i W_i}{\sum_{i \in \mathcal{K}} F_i}. \quad (5)$$

Here, $F_i$ acts as a weight that amplifies the influence of more important experts in the merging process. By normalizing the weights using the sum of Fisher importance values, we ensure that the merged base weight remains appropriately scaled and the delta weights $\Delta W_i$ are more compact and exhibit stronger low-rank properties. This facilitates the application of low-rank compression techniques, such as Singular Value Decomposition (SVD), in later stages of our framework (see Section 3.3). The improved compressibility of delta weights reduces both storage requirements and memory overhead during inference.

### 3.3. Truncation-aware Singular Value Decomposition

To compress the delta weights $\Delta W_i$, we apply the truncation-aware SVD approach in SVD-LLM (Wang et al., 2024) that enhances traditional decomposition methods by incorporating activation statistics. For each delta weight $\Delta W_i$, we first compute its activation-weighted representation, $W_{\text{scale}}$, as:

$$W_{\text{scale}} = \Delta W_i S_i, \quad (6)$$

where $S_i \in \mathbb{R}^{n \times n}$ is derived from the activation Gram matrix $X_i X_i^T$, with $X_i$ representing the i th expert's activation matrix. Specifically, $S_i$ is computed using Cholesky decomposition of the Gram matrix. Using $W_{\text{scale}}$, we perform SVD to decompose it into three components:

$$W_{\text{scale}} = U \Sigma V^T, \quad (7)$$

where $U \in \mathbb{R}^{m \times k}$ and $V \in \mathbb{R}^{n \times k}$ are orthogonal matrices, and $\Sigma \in \mathbb{R}^{k \times k}$ is a diagonal matrix containing the singular values. We then truncate the smallest singular values in $\Sigma$ to retain only the top-$k$ components, then the compressed delta U matrix and V matrix are as follows:

$$\begin{aligned}
\Delta U_i &= U \sqrt{\text{Trunc}(\Sigma)} \\
\Delta V_i &= \sqrt{\text{Trunc}(\Sigma)} V^T S_i^{-1}
\end{aligned} \quad (8)$$

This truncation-aware SVD approach mitigates reconstruction loss caused by activation outliers and ensures a mapping between singular values and compression loss to preserve the essential characteristics of the original weight distribution, enabling a more effective compression process.

## 3.4. Semi-dynamical Structured Pruning

The base weight matrix $W_b$ in our framework represents a combination of multiple expert weights, making them full-rank and highly expressive. However, their high dimensionality introduces significant redundancy, which increases storage and computational costs during inference. Traditional low-rank decomposition or static pruning methods often fail to effectively compress these base weights without incurring substantial performance degradation, owing to the unique structure of the base weights that stores information from all experts. Through empirical analysis, we observe a key phenomenon: while a subset of the columns in the base weights matrix consistently exhibits negligible contributions across different inputs (static redundancy), the relative importance of the remaining columns varies significantly depending on the input batch (dynamic redundancy). This insight motivates us to develop **new two-phase (first static then dynamic) pruning paradigm** that separately handles these two types of redundancies: **In the static pruning phase,** we identify and prune columns of $W_b$ that consistently contribute the least across all inputs. To achieve this, we compute a column-wise pruning metric that combines the magnitude of the weights and their interaction with the input activations. Specifically, the pruning metric for the $j$-th column of $W_b$ is computed as:

$$C_j = \|W_b[:,j]\|_2 \cdot \|X[j,:]\|_2, \tag{9}$$

where $W_b \in \mathbb{R}^{m \times n}$ with $m$ number of rows and $n$ number of columns (channels), $X \in \mathbb{R}^{n \times b}$ represent the input activations for a batch of size $b$. $\|W_b[:,j]\|_2$ is the $L_2$ norm of the $j$-th column of $W_b$, and $\|X[j,:]\|_2$ is the $L_2$ norm of the activations corresponding to the $j$-th column. We then sort all columns by their pruning metric $C_j$ in ascending order and prune the lowest-scoring columns to achieve half of the target sparsity level. **In the dynamic pruning phase,** we handle input-dependent redundancies by dynamically updating the pruning metrics for the remaining columns based on the current input batch. For a given batch of inputs $X$, we recompute the column-wise pruning metric: $C_j^{\text{dynamic}} = \|W_b[:,j]\|_2 \cdot \|X[j,:]\|_2$, but only for the columns retained after static pruning. We then prune the lowest-scoring columns to achieve the remaining half of the target sparsity. This dynamic pruning ensures that the model adapts to the specific input distribution of each batch, optimizing the number of active parameters during inference.

## 3.5. Overall Algorithm Procedure

The overall algorithm flow is summarized in Figure 3, which outlines the main steps of our framework, including Fisher-weighted merging of base weights, delta weight compression, and semi-dynamical structured pruning for base weights. In the forward pass, the process uses sparse gating to activate only the top-$k$ delta weights for each input. For example, the gating function selects the top-$k$ most relevant experts' delta weights based on $G(x)$, and their contributions are aggregated along with the shared base weight. The forward computation can be expressed as:

$$y = W_b x + \sum_{i=1}^{\mathcal{K}} G(x)_i \cdot \Delta U_i \Delta V_i \, x_{[selected\ token]}, \tag{10}$$

where $\Delta U_i$ and $\Delta V_i$ are the decomposed delta weights of selected experts. This structure ensures efficient sparse computation while leveraging the specialized knowledge of the selected experts.

**Parameter Compression Analysis.** For $n$ experts with $m$ the size of individual parameters, we assign $p\%$ the compression ratio for delta weights, and $s\%$ the compression ratio for the base weight after pruning. For **static parameter storage**, the original model requires $n \cdot m$ parameters for the experts. After delta decomposition, the storage requirement increases slightly to $(n+1)m$ due to the addition of the shared base weight $W_b$. After static compression, storage parameters can be expressed as: $(n \cdot p\% + s\%/2)m$, For **activation parameter reduction**, the original activation storage requirement is $k \cdot m$, as top-$k$ experts are active at a time. After compression, the activation parameter requirement becomes: $(k \cdot p\% + s\%)m$.

# 4. Experiments

In this section, we conduct a comprehensive series of experiments to evaluate the effectiveness of our proposed $D^2$-MoE method. We first compare our approach with state-of-the-art compression methods across various MoE models at different compression ratios. To provide deeper insights into our method's performance, we also conduct detailed ablation studies on $D^2$-MoE. All experiments are performed on NVIDIA A100 GPUs.

## 4.1. Experimental Setups

**Models and Datasets.** To showcase the versatility of our $D^2$-MoE method, we assess its effectiveness on common MoE models: Mixtral-8×7B, DeepSeek-moe-16b-base, Phi-3.5-MoE and Qwen2-57B-A14B. Mixtral-8×7B employs 8 experts, and Phi-3.5-MoE features 16 experts, each with 3.8 billion parameters. In comparison, DeepSeek-moe-16b-base and Qwen2-57B-A14B adopt an even more fine-grained expert architecture, leveraging 64 experts. We conduct experiments on MoE models with fewer experts, such as Mixtral-8x7B and Phi-3.5-MoE, as well as those with a greater number of experts, such as DeepSeekMoE-16B-Base and Qwen2-57B-A14B, to demonstrate the versatility of $D^2$-MoE. We evaluate our method across 10 datasets, encompassing 3 language modeling datasets (WikiText-

*Table 2.* Performance of $D^2$-MoE for Mixtral-8×7B, DeepSeekMoE-16B-Base, Phi-3.5-MoE and Qwen2-57B-A14B on 3 language modeling datasets (measured by perplexity ($\downarrow$)) and 7 common sense reasoning datasets (measured by accuracy ($\uparrow$)).

| Ratio | Method | WikiText-2↓ | PTB↓ | C4↓ | Openb. | ARC_e | WinoG. | HellaS. | ARC_c | PIQA | MathQA | Average↑ |
|---|---|---|---|---|---|---|---|---|---|---|---|---|
| | | | | | **Mixtral-8×7B** | | | | | | | |
| 0% | Original | 3.98 | 12.99 | 6.78 | 0.36 | 0.84 | 0.76 | 0.65 | 0.57 | 0.82 | 0.43 | 0.63 |
| 20% | NAEE (2024b) | 4.77 | 16.09 | 8.89 | 0.32 | 0.76 | 0.72 | 0.58 | 0.47 | 0.79 | 0.40 | 0.58 |
| | MoE-I$^2$ (2024) | 4.86 | 26.50 | 11.07 | 0.32 | 0.79 | 0.74 | 0.55 | 0.48 | 0.78 | 0.37 | 0.57 |
| | $D^2$-MoE (Ours) | **4.65** | **16.32** | **8.59** | **0.33** | **0.80** | **0.75** | **0.61** | **0.51** | **0.81** | **0.39** | **0.60** |
| 40% | NAEE (2024b) | 6.44 | 22.15 | 13.86 | 0.25 | 0.63 | 0.64 | 0.46 | 0.36 | 0.72 | **0.35** | 0.48 |
| | MoE-I$^2$ (2024) | 6.74 | 60.45 | 22.44 | 0.26 | 0.71 | 0.66 | 0.43 | 0.38 | 0.69 | 0.31 | 0.49 |
| | $D^2$-MoE (Ours) | **5.28** | **20.54** | **10.10** | **0.32** | **0.78** | **0.73** | **0.57** | **0.47** | **0.78** | 0.34 | **0.57** |
| 60% | NAEE (2024b) | 11.43 | 47.28 | 31.16 | 0.17 | 0.42 | 0.55 | 0.33 | 0.23 | 0.62 | 0.26 | 0.36 |
| | MoE-I$^2$ (2024) | 13.52 | 182.99 | 74.62 | 0.18 | 0.44 | 0.55 | 0.32 | 0.22 | 0.58 | 0.23 | 0.36 |
| | $D^2$-MoE (Ours) | **6.46** | **23.63** | **12.76** | **0.28** | **0.72** | **0.71** | **0.51** | **0.38** | **0.73** | **0.31** | **0.52** |
| | | | | | **DeepSeekMoE-16B-Base** | | | | | | | |
| 0% | Original | 6.38 | 9.47 | 9.82 | 0.32 | 0.76 | 0.70 | 0.58 | 0.44 | 0.79 | 0.31 | 0.56 |
| 20% | NAEE (2024b) | 9.44 | 15.02 | 15.34 | **0.32** | 0.71 | 0.66 | 0.55 | 0.40 | **0.77** | 0.29 | 0.53 |
| | MoE-I$^2$ (2024) | 7.69 | 11.59 | 13.72 | 0.26 | 0.71 | 0.68 | 0.49 | 0.38 | 0.73 | 0.29 | 0.50 |
| | $D^2$-MoE (Ours) | **6.84** | **11.10** | **11.88** | 0.30 | **0.74** | **0.69** | **0.55** | **0.41** | 0.76 | **0.31** | **0.54** |
| 40% | NAEE (2024b) | 8.55 | 14.47 | 17.98 | 0.23 | 0.67 | **0.67** | 0.41 | 0.32 | 0.69 | 0.26 | 0.46 |
| | MoE-I$^2$ (2024) | 9.73 | 15.75 | 19.75 | 0.23 | 0.64 | 0.66 | 0.41 | 0.31 | 0.68 | 0.26 | 0.45 |
| | $D^2$-MoE (Ours) | **7.93** | **14.07** | **15.18** | **0.26** | **0.69** | 0.65 | **0.45** | **0.36** | **0.72** | **0.28** | **0.49** |
| 60% | NAEE (2024b) | 23.20 | 49.89 | 48.63 | 0.17 | 0.49 | 0.58 | 0.33 | 0.24 | 0.61 | 0.23 | 0.38 |
| | MoE-I$^2$ (2024) | 15.83 | 32.2 | 38.60 | 0.17 | 0.48 | 0.58 | 0.32 | 0.23 | 0.61 | 0.22 | 0.37 |
| | $D^2$-MoE (Ours) | **11.67** | **27.73** | **27.63** | **0.21** | **0.54** | **0.61** | **0.35** | **0.29** | **0.63** | **0.24** | **0.41** |
| | | | | | **Phi-3.5-MoE** | | | | | | | |
| 0% | Original | 3.48 | 8.43 | 8.22 | 0.40 | 0.77 | 0.76 | 0.68 | 0.56 | 0.79 | 0.38 | 0.62 |
| 40% | NAEE (2024b) | 8.18 | 20.07 | 16.11 | **0.35** | **0.73** | 0.73 | 0.61 | 0.48 | 0.76 | 0.37 | 0.57 |
| | MoE-I$^2$ (2024) | 7.46 | 20.95 | 20.95 | 0.29 | 0.59 | 0.67 | 0.27 | 0.40 | 0.70 | 0.25 | 0.45 |
| | $D^2$-MoE (Ours) | **6.07** | **13.79** | **14.01** | 0.34 | 0.72 | **0.73** | **0.65** | **0.53** | **0.78** | **0.38** | **0.60** |
| | | | | | **Qwen2-57B-A14B** | | | | | | | |
| 0% | Original | 5.12 | 9.18 | 8.86 | 0.33 | 0.75 | 0.74 | 0.63 | 0.46 | 0.81 | 0.39 | 0.59 |
| 40% | NAEE (2024b) | **6.81** | 11.34 | **11.57** | 0.31 | 0.73 | 0.73 | 0.55 | **0.46** | 0.76 | 0.36 | 0.55 |
| | MoE-I$^2$ (2024) | 24.90 | 77.05 | 22.50 | 0.26 | 0.70 | 0.46 | 0.71 | 0.41 | 0.75 | 0.30 | 0.51 |
| | $D^2$-MoE (Ours) | 8.19 | **11.23** | 12.70 | **0.33** | **0.75** | **0.75** | **0.61** | 0.45 | **0.79** | **0.36** | **0.58** |

2 (Merity et al., 2017), PTB (Marcus et al., 1993), and C4 (Raffel et al., 2020)), along with 7 common sense reasoning datasets (OpenbookQA (Mihaylov et al., 2018), WinoGrande (Sakaguchi et al., 2020), HellaSwag (Zellers et al., 2019), PIQA (Bisk et al., 2020), MathQA (Amini et al., 2019), ARC-e, and ARC-c (Clark et al., 2018)) in a zero-shot setting using the LM-Evaluation-Harness framework (Gao et al., 2023).

**Implementation Details.** For fair comparisons, we use 512 random samples from WikiText-2 as calibration data to conduct all experiments. We focus on compressing the model without retraining the full model parameters. See Appendix A for more details.

### 4.2. Compression Performance and Comparisons

**Main Results in Multiple MoE LLMs.** Our experimental results shown in Table 2 demonstrate the superior performance of $D^2$-MoE across different MoE models and compression ratios. On Mixtral-8×7B, at 20% compression, $D^2$-MoE achieves an average score of 0.60 (95.2%

of the original performance 0.63), outperforming NAEE (0.58) and MoE-I$^2$ (0.57). Even at 60% compression, our method maintains a competitive score of 0.52, significantly surpassing both baselines (0.36). The advantages extend to models with more experts. For DeepSeek-MoE-16B-Base, our method achieves average scores of 0.54, 0.49, and 0.41 at 20%, 40%, and 60% compression respectively, showing significant improvements over baselines, particularly in perplexity metrics. Similar superior performance is observed on Phi-3.5-MoE and Qwen2-57B-A14B, demonstrating the effectiveness of $D^2$-MoE across different model scales. More detail results are shown in Appendix A.1.

**Comparing to Different Compressors.** We compare $D^2$-MoE against three categories of compression methods on Mixtral-8x7B at 20% compression ratio: pruning-based methods (SparseGPT, NAEE), SVD-based methods (ASVD, MoE-I$^2$), and hybrid methods (LoSparse, MC-SMoE, MoE-Compress). As shown in Table 3, $D^2$-MoE achieves the best overall performance with an average score of 0.60 (95.2% of original 0.63), outperforming all baselines across multiple metrics. Our method shows competitive perplexity scores

*Table 3.* Performance of Mixtral-8×7B compressed by $D^2$-MoE under 20% compression ratios.

| Methods | WikiText-2↓ | PTB↓ | C4↓ | Openb. | ARC_e | WinoG. | HellaS. | ARC_c | PIQA | MathQA | Average↑ |
|---|---|---|---|---|---|---|---|---|---|---|---|
| Original | 3.98 | 12.99 | 6.78 | 0.36 | 0.84 | 0.76 | 0.65 | 0.57 | 0.82 | 0.43 | 0.63 |
| NAEE (2024b) | 4.77 | 16.09 | 8.89 | 0.32 | 0.76 | 0.72 | 0.58 | 0.47 | 0.79 | 0.40 | 0.58 |
| SparseGPT(2:4) (2023) | 4.69 | 21.11 | 9.19 | 0.30 | 0.77 | 0.74 | 0.56 | 0.45 | 0.77 | 0.35 | 0.56 |
| MoE-I$^2$ (2024) | 4.86 | 26.50 | 11.07 | 0.32 | 0.79 | 0.74 | 0.55 | 0.48 | 0.78 | 0.37 | 0.57 |
| ASVD (2023b) | 9.44 | 47.29 | 20.30 | 0.25 | 0.71 | 0.66 | 0.48 | 0.40 | 0.73 | 0.35 | 0.51 |
| LoSparse (2023d) | 953.51 | 805.16 | 1273.12 | 0.20 | 0.27 | 0.49 | 0.28 | 0.26 | 0.53 | 0.20 | 0.32 |
| MC-SMoE (2024f) | 1341.36 | 1316.52 | 1478.13 | 0.26 | 0.28 | 0.51 | 0.29 | 0.25 | 0.54 | 0.19 | 0.33 |
| MoE-Compress (2024) | 6.12 | 14.67 | 11.61 | 0.30 | 0.73 | 0.70 | 0.54 | 0.46 | 0.73 | 0.33 | 0.54 |
| $D^2$-MoE | **4.65** | **16.32** | **8.59** | **0.33** | **0.80** | **0.75** | **0.61** | **0.51** | **0.81** | **0.39** | **0.60** |

*Table 4.* Throughput (Tokens/sec), Memory of Mixtral-8x7B model under 60%, 70%, 80% compress ratio of different methods. And the perplexity on WikiText-2 of 60%, 70%, 80% is 6.35, 8.15, 12.95 respectively which gain well performance under high compression ratio.

| Methods | BSZ=64, Ratio=60% | | BSZ=64, Ratio=70% | | BSZ=64, Ratio=80% | |
|---|---|---|---|---|---|---|
| Model Size | 18.68B | | 14.01B | | 9.33B | |
| Memory | 34.8G | | 26.1G | | 17.3G | |
| | TFLOPs | Tokens/sec | TFLOPs | Tokens/sec | TFLOPs | Tokens/sec |
| NAEE | 481 | 271.89 | 386 | 272.66 | 290 | 278.53 |
| MoE-I$^2$ | 838 | 227.60 | 743 | 252.55 | 647 | 294.04 |
| LoSparse | 1150 | 158.90 | 1240 | 191.45 | 1330 | 198.04 |
| $D^2$-MoE | 481 | 277.72 | 386 | 300.33 | 290 | 313.29 |

*Table 5.* Perplexity of different merge methods for Base Weights.

| Method | WikiText-2↓ | PTB↓ | C4↓ |
|---|---|---|---|
| Original | 3.98 | 12.99 | 6.78 |
| Mean average | 7.66 | 46.85 | 24.39 |
| TIES (2024) | 12.45 | 87.31 | 29.10 |
| RegMean (2022) | 187.19 | 1206.05 | 612.70 |
| Frequency (2023c) | 6.42 | 35.12 | 13.79 |
| **Fisher (Ours)** | 5.28 | 20.54 | 10.10 |

*Table 6.* Perplexity of compressors for Base and Delta Weights.

| Part | Method | WikiText-2↓ | PTB↓ | C4↓ |
|---|---|---|---|---|
| $W_b$ | Truncation-aware SVD | 5.63 | 27.40 | 12.65 |
| | Static Pruner | 5.31 | 20.43 | 10.75 |
| | Semi-dynamic Pruner | 5.28 | 20.54 | 10.10 |
| $\Delta W_i$ | Pruning | 5.74 | 20.83 | 11.48 |
| | Vanilla SVD | 6.22 | 22.54 | 10.72 |
| | Activation-aware SVD | 5.91 | 22.63 | 11.31 |
| | Truncation-aware SVD | 5.28 | 20.54 | 10.10 |

performing NAEE by 12.5% and MoE-I$^2$ by 6.5%. In contrast, LoSparse uses 4.6× more TFLOPs (1330) but achieves only 198.04 tokens/sec. Meanwhile, our method maintains reasonable perplexity on WikiText-2 (6.35, 8.15, and 12.95 at 60%, 70%, and 80% compression), demonstrating an optimal balance between efficiency and performance. Refer to Appendix A.3 to see detail results analysis.

### 4.3. Ablation study

**Various Merger for Base Weights.** Table 5 delves into the performance of different merge methods for base weights, demonstrate the effectiveness of our Fisher merge (Matena & Raffel, 2022) method compared to other base weight merging techniques, including mean average, expert frequency average (Li et al., 2023c), RegMean (Jin et al., 2022), and TIES (Yadav et al., 2024). The analysis reveals that while expert frequency merging which uses expert activation frequency for weighted averaging on base weights demonstrates promising results, our Fisher merge method selectively extracts important weights from different experts and integrates them into the base weights which achieves the lowest perplexity scores on WikiText-2 (5.28), PTB (20.54), and C4 (10.10) on Mixtral-8x7B under 40% compression ratios setting.

**Varying Compressors for Base and Delta Weights.** Table 6 unveils the different methods to compress both base and delta weights. For base weights, our dynamic pruning method achieves superior performance, with the lowest perplexity scores on the WikiText-2 (5.28), PTB (20.60), and C4 (10.12)datasets, outperforming both truncation-aware

on WikiText-2 (4.65), PTB (16.32), and C4 (8.59), surpassing both pruning and SVD-based methods. For downstream tasks, $D^2$-MoE achieves superior performance on reasoning tasks like ARC-e (0.80) and WinoG (0.75). While hybrid methods show significant degradation, especially in perplexity metrics, our method maintains consistent performance across all evaluations, demonstrating an optimal balance between compression and model capabilities. See Appendix A.2 to get more detail results analysis.

**Inference Speed Acceleration and Memory Reduction.** Table 4 demonstrates the inference efficiency of various methods on Mixtral-8×7B (batch size=64) under high compression settings. At 60% compression, $D^2$-MoE achieves 277.72 tokens/sec with only 481 TFLOPs, surpassing NAEE in throughput while matching its computational efficiency. MoE-I$^2$ and LoSparse require significantly more TFLOPs (838 and 1150) but deliver lower throughput. The advantages become more evident at 80% compression, where $D^2$-MoE achieves 313.29 tokens/sec with 290 TFLOPs, out-

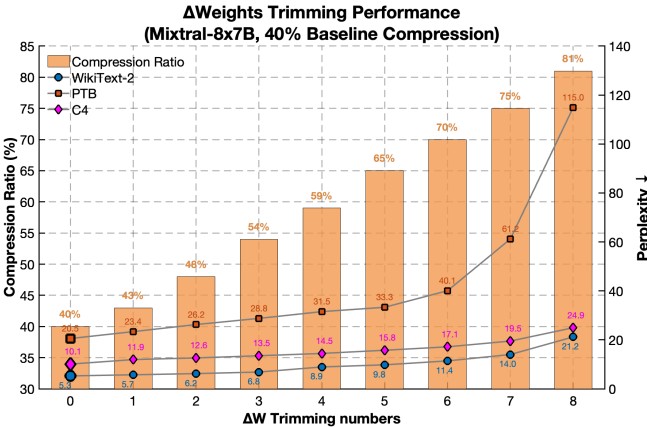

*Figure 4.* Expanding $D^2$-MoE via Delta Weights Trimming.

*Table 7.* Perplexity of various individual ratios for Base and Delta Weights on Mixtral-8x7B 40% compression ratios.

| $W_b$ | $\Delta W_i$ | **WikiText-2↓** | **PTB↓** | **C4↓** |
|---|---|---|---|---|
| 10% | 52.66% | 5.63 | 27.40 | 12.65 |
| 20% | 51.41% | 5.31 | 20.43 | 10.75 |
| 30% | 50.16% | 5.28 | 20.54 | 10.10 |
| 40% | 48.90% | 5.74 | 20.83 | 11.48 |
| 50% | 47.67% | 6.22 | 22.54 | 10.72 |
| 60% | 46.42% | 5.28 | 20.54 | 10.10 |

SVD with a scale matrix using in SVD-LLM and static pruning (Wanda-sp). For delta weights, we compare pruning, Vanilla SVD without scale matrix, activation-aware SVD (Yuan et al., 2023a). Our truncation-aware SVD method with scale matrix achieves the best performance. Finally, we use semi-dynamic pruning for base weights and truncation-aware SVD for delta weights in our $D^2$-MoE.

**Sensitivity of Compression Ratio Hyperparameters.** Table 7 demonstrate the sensitivity of compression ratio between base weights and delta weights. Under the setting of 40% compression of Mixtral-8x7B model, we observe that less compression of the base weights generally leads to better performance, as it preserves more critical information among all experts and maintains model accuracy. However, there is a trade-off between the compression ratio and inference time speedup, as higher compression ratios of base weights typically result in faster inference but may degrade performance. After careful evaluation, we choose a balanced ratio that optimizes both performance and inference efficiency, and more Hyperparameters are in Appendix B.5.

**Expanding $D^2$-MoE via Delta Weights Trimming.** Figure 4 shows how delta weights trimming affects $D^2$-MoE's performance. As trimming increases, perplexity rises grad-

*Table 8.* Perplexity of 40% compressed Mixtral-8×7B via calibration data with varying number from WikiText-2 and C4

| Method | 32 | 64 | 128 | 256 | 512 |
|---|---|---|---|---|---|
| WikiText-2↓ | 5.81 | 5.56 | 5.48 | 5.37 | 5.28 |
| C4↓ | 10.92 | 10.79 | 10.65 | 10.52 | 10.10 |

*Table 9.* Perplexity of varying calibration data on Mixtral-8x7B 40% compression ratios.

| **Calibration** | **WikiText-2↓** | **PTB↓** | **C4↓** |
|---|---|---|---|
| Wikitext-2 | 5.28 | 20.54 | 10.10 |
| C4 | 5.37 | 21.03 | 11.52 |

ually, but our method maintains competitive performance across different compression ratios (43%-81%). With 1 trimming delta weight (43% compression), we achieve a WikiText-2 perplexity of 6.43, comparable to non-trim models. Even with 7 trimming experts (75% compression), the perplexity remains reasonable at 14.71, demonstrating effective balance between compression and performance. More detailed results can be found in Appendix B.1.

**Impact of Calibration Data.** Table 9 demonstrates that the choice of calibration data, whether WikiText-2 or C4, has minimal influence on overall task performance, highlighting the robustness of our method across diverse datasets. Table 9 explores the effects of varying the number of calibration samples. Results indicate that increasing the number of data samples generally leads to a decrease in perplexity, suggesting improved performance with more samples.

## 5. Conclusion

In this work, we present $D^2$-MoE, a unified framework for compressing MoE LLMs by addressing the inherent redundancy in their weight structures. Our approach systematically integrates delta decomposition, Fisher-weighted merging, truncation-aware singular value decomposition (SVD), and semi-dynamical structured pruning to achieve efficient parameter reduction while maintaining the performance of MoE models. By decomposing expert weights into a shared base weight and expert-specific delta weights, we effectively isolate common structures and reduce redundancy. Our empirical analysis demonstrates that $D^2$-MoE achieves significant parameter compression while preserving the predictive performance of MoE models on benchmark tasks. Future work may explore integrating $D^2$-MoE with advanced training techniques, such as knowledge distillation and parameter quantization. We hope that the proposed framework contributes to the broader field of efficient large-scale modeling, offering a practical pathway for deploying high-capacity MoE models in real-world applications.

**Limitations** Our $D^2$-MoE involves decomposition and pruning steps with some complexity (see more analysis in Appendix A.4). We aim to simplify it in future work.

## Acknowledgements

This work is funded in part by the HKUST Start-up Fund (R9911), Theme-based Research Scheme grant (T45-205/21-N), the InnoHK funding for Hong Kong Generative AI Research and Development Center, Hong Kong SAR, and the research funding under HKUST-DXM AI for Finance Joint Laboratory (DXM25EG01).

## Impact Statement

The primary focus of this work is to develop and evaluate technical approaches for improving the storage and reasoning efficiency of MoE LLMs. By addressing the inherent redundancy in MoE architectures, our $D^2$-MoE framework contributes to the ongoing effort to design green and easy-to-use LLMs. All evaluations and experiments are performed on publicly available benchmarks, ensuring transparency and reproducibility. We believe that our framework will not be controversial in ethical impacts and expected societal implications.

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

# Appendix

This appendix provides additional details and analyses to complement the experiments and methodology described in the main paper. We first present an extended discussion of experimental results, including comparisons with other methods, analyses of compression efficiency, and the impact of the calibration dataset. Next, we provide detailed information about our experimental setups, including the evaluated Mixture-of-Experts (MoE) models, datasets, hyperparameters, and experimental configurations. Finally, we include algorithmic tables and pseudo-code for key components of the $D^2$-MoE framework to ensure clarity and reproducibility of our approach.

# A. More Discussion and Experimental Results

## A.1. Detail Analysis of Main Results

Our experimental results demonstrate the superior performance of $D^2$-MoE across different MoE models and compression ratios. On Mixtral-8×7B, our method consistently outperforms existing approaches across all compression ratios (20%, 40%, and 60%). At 20% compression, $D^2$-MoE achieves an average score of 0.60, maintaining 95.2% of the original model's performance (0.63), while NAEE and MoE-I$^2$ only achieve 0.58 and 0.57 respectively. Notably, even at aggressive 60% compression, $D^2$-MoE maintains a competitive average score of 0.52, significantly surpassing NAEE (0.36) and MoE-I$^2$ (0.36). The advantages of $D^2$-MoE are further validated on models with more experts. For DeepSeek-MoE-16B-Base, our method maintains stable performance across different compression ratios, achieving average scores of 0.54, 0.49, and 0.41 at 20%, 40%, and 60% compression respectively. This represents a significant improvement over baseline methods, particularly in perplexity metrics (WikiText-2↓, PTB↓, and C4↓) where our method shows orders of magnitude better results compared to MoE-I$^2$. Similar patterns are observed in Phi-3.5-MoE and Qwen2-57B-A14B, where $D^2$-MoE consistently maintains higher performance scores while achieving target compression ratios. Most remarkably, our method exhibits exceptional stability in maintaining model performance across different evaluation tasks. For instance, on downstream tasks such as ARC-e, WinoG, and PIQA, $D^2$-MoE consistently preserves close to 90% of the original model's performance at 20% compression across all tested models. This demonstrates that our compression method not only achieves high compression ratios but also preserves the model's general language understanding and reasoning capabilities.

## A.2. Comparison with Other Methods

Our experimental results demonstrate the superior effectiveness of $D^2$-MoE in compressing the Mixtral-8x7B model at a 20% compression ratio, outperforming various state-of-the-art compression methods across multiple metrics, as shown in Table 3. Specifically, we compare our method against three categories of compression approaches: (1) pruning-based methods (SparseGPT, NAEE), (2) SVD-based methods (ASVD, MoE-I$^2$), and (3) hybrid methods that combine multiple compression techniques (LoSparse, MC-SMoE, MoE-Compress). $D^2$-MoE achieves the best overall performance with an average score of 0.60, maintaining 95.2% of the original model's capabilities (0.63). In perplexity evaluations, our method achieves competitive scores on WikiText-2 (4.65), PTB (16.32), and C4 (8.59), matching or outperforming pure pruning methods like NAEE and SparseGPT. Notably, our approach significantly outperforms SVD-based methods such as ASVD and MoE-I$^2$, which achieve average scores of 0.51 and 0.57 respectively. For downstream tasks, $D^2$-MoE demonstrates remarkable performance, particularly in reasoning tasks such as ARC-e (0.80) and WinoG (0.75), surpassing all baseline methods. The hybrid methods (LoSparse and MC-SMoE) show significant degradation in performance, especially in perplexity metrics, while our method maintains stable performance across all evaluation dimensions. Other model compression methods follow different technical approaches—targeting either dense models (Dong et al., 2024; Li et al., 2024c;g; uyuk et al., 2024), smaller architectures (Li et al., 2024d;e), or addressing other aspects of compression such as quantization (Dong et al., 2025b; 2023b), AutoML (Dong et al., 2025a; 2023a), or distillation (Li et al., 2024a;b; 2023a; Li & Jin, 2022; Li, 2022).

This comprehensive comparison validates that $D^2$-MoE effectively preserves model capabilities while achieving the desired compression ratio, striking an optimal balance between model efficiency and performance.

## A.3. Analysis of Runtime SpeedUp and Memory Usage

Table 4 demonstrates significant hardware inference acceleration across high compression ratios. We evaluate the inference efficiency of various methods on Mixtral-8×7B with a batch size of 64 under high compression settings (60%, 70%, and 80%). Our $D^2$-MoE achieves superior throughput while maintaining the lowest TFLOPs among all compared methods. At 60% compression (18.68B parameters, 34.8G memory), $D^2$-MoE achieves 277.72 tokens/sec throughput while requiring

only 481 TFLOPs, matching NAEE's computational efficiency but with 2.1% higher throughput. In contrast, MoE-I$^2$ and LoSparse require substantially higher computational resources (838 and 1150 TFLOPs respectively) while delivering lower throughput. The efficiency advantages of $D^2$-MoE become more pronounced at higher compression ratios. At 80% compression (9.33B parameters, 17.3G memory), our method achieves 313.29 tokens/sec, outperforming NAEE by 12.5% and MoE-I$^2$ by 6.5% in throughput while maintaining the lowest TFLOPs (290). Notably, LoSparse, despite using 4.6× more TFLOPs (1330), achieves only 198.04 tokens/sec, demonstrating the superior efficiency of our approach. While achieving these significant speedups, $D^2$-MoE maintains reasonable model performance. The perplexity scores on WikiText-2 at 60%, 70%, and 80% compression ratios are 6.35, 8.15, and 12.95 respectively, showing a gradual and controlled degradation even at extreme compression levels. These results highlight that $D^2$-MoE not only achieves better compression quality but also delivers practical benefits in terms of inference speed and computational efficiency, making it particularly attractive for real-world deployments where both model size and inference speed are critical considerations.

### A.4. Computational Cost Discussion of $D^2$-MoE

We present a detailed computational cost analysis for each stage of our compressing procedure. The $D^2$-MoE merging approach encompasses two principal stages: 1.Metric calculate and 2.Merging expert weights. To begin with, the Fisher metric is calculated by feeding a calibration dataset through the model and computing gradients with respect to each weight parameter. Secondly, Expert weights merging is performed through Fisher-weighted averaging, where the Fisher metric of each parameter serves as its importance weight in the merging process. This Process spends 11mins when conducting on Mixtral-8x7B. Then, we compute the scale matrix for SVD and decompose the delta weights using Singular Value Decomposition (SVD). In terms of computational cost, collecting the scale matrix through calibration takes 19 minutes, while performing SVD decomposition on delta weights requires 25 minutes on Mixtral models. These two steps constitute the main computational overhead of our method. Time of other models are shown in Table 10.

*Table 10.* Compressing time and memory used of different models

| Stage | Metric | Mixtral-8x7B | DeepSeekMoE-16B-Base | Phi3.5-MoE | Qwen2-57B-A14B |
|---|---|---|---|---|---|
| Merging | Time Cost | 11 mins | 8 mins | 13 mins | 23 mins |
| | Memory Cost | 131.01 GB | 34.6 GB | 118.27 GB | 127.21 GB |
| SVD | Time Cost | 44 mins | 15 mins | 35 mins | 59 mins |
| | Memory Cost | 109.45 GB | 52.9 GB | 117.41 GB | 126.41 GB |

### A.5. Additional Results on Compression Ratios between Base Weights and Delta Weights

In Table 7, we provide additional results for Mixtral-8x7B across a range of compression ratios. These results highlight the flexibility of $D^2$-MoE in balancing compression and performance. For example, at the 40% compression ratio, $D^2$-MoE achieves a WikiText-2 perplexity of **5.28**, significantly lower than competing methods. We ultimately selected a 10% pruning ratio for the Base weights and a corresponding SVD decomposition ratio for the Delta weights to preserve performance, as the Base weights contain more shared knowledge across all experts. Additionally, a 60% pruning ratio for the Base weights is chosen when prioritizing throughput efficiency.

### A.6. Pushing the Limit of Compressing Delta Weight

Based on the CKA (Centered Kernel Alignment) similarity analysis demonstrated in Figure 5 of delta weights among V matrices and U matrices in the Mixtral-8x7B model, we observe that all V matrices exhibit extremely high CKA similarity scores of approximately 0.9. This significant finding motivates us to share a single V matrix among experts to further compress the delta weights. Furthermore, the U matrices demonstrate moderate CKA similarity scores around 0.3, indicating considerable redundancy among them. This observation suggests that we can merge the U matrices in a manner similar to base weight compression(*e.g.*fisher merge), thereby achieving additional parameter reduction in U matrices. Refer to Table 11 to see the results.

### A.7. Adaptive Compression Ratio for Delta Weight

Given the observed layer-wise sensitivity shown in Figure6 in the importance of delta weights, we propose an adaptive compression strategy that allocates more parameters to sensitive layers while maintaining the same total parameter budget. As illustrated in Figure 6, our analysis reveals significant variations in layer sensitivity, with layer 2 exhibiting the highest

| Methods | WikiText-2↓ | PTB↓ | C4↓ | Tokens/sec |
|---|---|---|---|---|
| ShareV 80% ratio | 12.95 | 33.30 | 17.14 | 337.82 |
| MergeU 80% ratio | 13.21 | 40.07 | 19.50 | 290.55 |
| ShareV+MergeU 85% ratio | 18.93 | 61.18 | 24.94 | 328.04 |

*Table 11.* Perplexity and Throughput of ShareV and MergeU methods on extremly high compression ratio

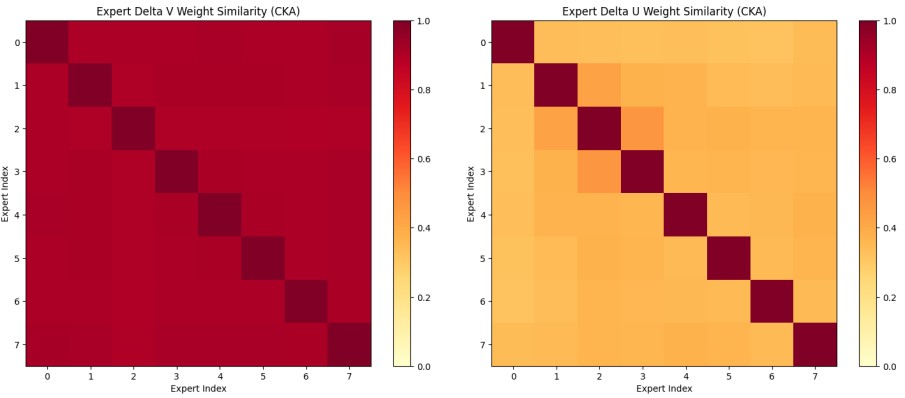

*Figure 5.* CKA of Delta V weights and Delta U weights of Mixtral-8x7B

sensitivity to compression while layer 1 showing the least sensitivity. Based on these findings, we implement an adaptive parameter allocation strategy where layer 2 receives the largest parameter budget, and layer 1 is assigned the smallest share of parameters, optimizing the distribution of compression ratios according to layer-wise sensitivity. This approach optimizes the distribution of parameters across layers based on their relative importance, potentially leading to better performance compared to uniform compression across all layers.

## B. More Detailed Experimental Details

### B.1. Details of Delta Weights Trimming

Figure 4 provides insights into the impact of delta weights trimming of $D^2$-MoE's performance. We utilize expert frequency as the criterion for trimming decisions, every time the lowest router sampling frequency expert's delta weight is trimed. The results show a general trend of raising perplexity as the number of trimmed delta weights increases, indicating a trade-off between model size reduction and performance retention. However, our method achieves significant compression ratios, ranging from 43% to 81%, while maintaining relatively low perplexity scores compared to baseline methods. For instance, with 1 trimming delta weight, we achieve a 43% compression ratio with a WikiText-2 perplexity of 6.43, which is competitive with non-trim compressed model. Even at higher compression ratios, such as 75% with 7 trimming experts, our method maintains a reasonable perplexity of 14.71 on WikiText-2, demonstrating its robustness. This highlights the advantage of our approach in balancing model efficiency and performance, making it suitable for resource-constrained environments without substantial degradation in language modeling quality.

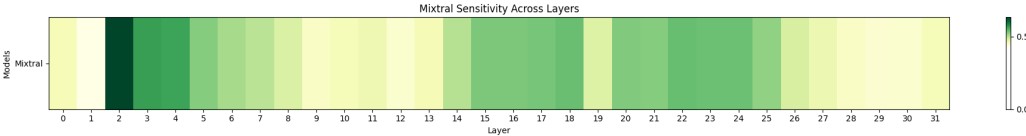

*Figure 6.* layer wise sensitivity of Mixtral-8x7B

## B.2. Motivation of CKA Similarity between experts and Single Value energy Retention

The CKA Similarity between experts reflects the redundancy among experts, which motivates us to merge their base weights to capture shared knowledge, as formulated in Equation 11. Energy-Retention quantifies the amount of information preserved after singular value truncation in the SVD decomposition process. As formulated in Equation 12, where k represents the number of largest singular values retained, this metric helps us evaluate the effectiveness of our compression while maintaining essential information.

$$\text{CKA}(W_1, W_2) = \frac{\text{tr}(HKHL)}{\sqrt{\text{tr}(HKHK)\text{tr}(HLHL)}} \tag{11}$$

$$\text{Energy-Retention} = \frac{\sum_{i=1}^{k} \lambda_i}{\sum_{i=1}^{n} \lambda_i}, \quad \text{where} \quad \lambda_i = \sigma_i^2 \tag{12}$$

## B.3. Evaluated MoE Models and Datasets

We evaluate $D^2$-MoE on a range of common MoE large language models (LLMs), including:

- **Mixtral-8x7B**: A state-of-the-art MoE model developed by Mistral AI, featuring 8 experts with each expert containing 7 billion parameters represent large experts MoE model in our experiment. It employs a top-2 routing mechanism where two experts are activated for each token, demonstrating strong performance across various tasks while maintaining computational efficiency.

- **DeepSeekMoE-16B-Base**: An advanced MoE architecture with 16 experts and a shared expert, where each expert contains 1 billion parameters represent small and fine-grained experts MoE model in our experiment. The model incorporates a shared expert mechanism to capture common knowledge across tasks, while specialized experts focus on domain-specific features. Its routing strategy dynamically selects the most relevant experts for each input token.

- **Phi-3.5-MoE**: Developed by Microsoft, this model features 16 experts, each containing 3.5 billion parameters. It implements a unique expert configuration that balances model capacity and computational efficiency. The model utilizes a sophisticated routing mechanism to effectively distribute computational load across experts.

- **Qwen2-57B-A14B**: A large-scale MoE model developed by Alibaba, comprising 64 experts and a shared expert, with a total of 57 billion parameters (14B active). The model employs an advanced sparse gating mechanism that activates only a small subset of experts for each token, achieving remarkable parameter efficiency despite its scale. The shared expert serves as a knowledge hub while specialized experts capture domain-specific features.

These models represent different design philosophies in MoE architecture, varying in their number of experts, parameter distribution, and routing strategies, providing a comprehensive testbed for evaluating our compression method.

We test these models on 10 datasets, including 3 language modeling datasets (WikiText-2, PTB, and C4) and 7 common-sense reasoning datasets (OpenbookQA, WinoGrande, HellaSwag, PIQA, MathQA, ARC-easy, and ARC-challenge).

## B.4. Calibration Dataset

We use WikiText-2 as the primary calibration dataset, selecting 512 random samples for all experiments. To assess the effect of calibration data type, we also use samples from C4. As shown in Table 9, WikiText-2 consistently outperforms C4 in terms of compression quality. The calibration data size plays an important role in our method's performance. While increasing calibration samples generally improves results, we observe that using 2048 samples leads to a slight performance degradation. Therefore, considering the trade-off between performance and computational efficiency, we choose 512 samples as our optimal calibration set size.

## B.5. Hyperparameters and Experimental Configurations

Table 12 lists the key hyperparameters used in our experiments, including the target compression ratios, SVD truncation thresholds, and sparsity levels for static and dynamic pruning.

*Table 12.* Hyperparameter Settings for $D^2$-MoE Experiments.

| Hyperparameter | 20% ratio | 40% ratio | 60% ratio | 70% ratio | 80% ratio |
|---|---|---|---|---|---|
| Pruning ratio for Performance | 10% of Base weights | | | | |
| Pruning ratio for Throughput | 60% of Base weights | | | | |
| SVD Truncation Threshold for Performance | preserve 68.05% | preserve 47.34% | preserve 26.62% | preserve 16.26% | preserve 5.93% |
| SVD Truncation Threshold for Throughput | preserve 74.30% | preserve 53.58% | preserve 32.86% | preserve 22.54% | preserve 12.18% |
| Static Pruning Sparsity for Performance | 5% of Base weights | | | | |
| Dynamic Pruning Sparsity for Performance | 5% of Base weights | | | | |
| Static Pruning Sparsity for Throughput | 30% of Base weights | | | | |
| Dynamic Pruning Sparsity for Throughput | 30% of Base weights | | | | |
| Calibration Dataset Size | 512 samples | | | | |
| Batch Size | 128 | | | | |

## C. Algorithm Tables and Pseudo-Code

We provide pseudo-code for key components of $D^2$-MoE to ensure reproducibility.

### C.1. Pytorch like Implement of D2MoE layer

### C.2. Pytorch like Implement of SVD and Merge Process

---

**Algorithm 2** Base Weight Merge and Delta SVD Decomposition for D2MoE

---

```python
class D2_MoE_Block(nn.Module):
    # input original experts weights, fisher_info, scale matrix
    def merge(original_experts, fisher_info, scale_matrix)

        # merge Base weight with fisher information
        weighted_sum = sum(fisher_info[idx] * original_experts[idx].weight for idx in range(self.num_experts))
        self.Base_W = weighted_sum / sum(fisher_info[idx] for idx in range(self.num_experts))

        for idx in range(self.num_experts):
            # calculate Delta weight
            Delta_W = original_experts[idx].weight - self.Base_W
            # decomposite Delta weight with Truncation-Aware SVD
            U, V = SVD(Delta_W, scale_matrix[idx])
            self.D2_experts[idx].Delta_U, self.D2_experts[idx].Delta_V = U, V

    def SVD(W, scale):
        # Truncation-Aware scale
        W_scale = W * scale

        inv_scale = torch.linalg.inv(scale)
        U, sigma, V = torch.linalg.svd(W_scale)

        truc_sigma = torch.diag(sigma[: num_trunc])
        svd_u = U[: num_trunc] * torch.sqrt(truc_sigma)

        # absorb scale into V
        svd_v = torch.sqrt(truc_sigma) * V[num_trunc :] * inv_scale
    return svd_u, svd_v
```

---

**Algorithm 1** D2MoE layer Implement

```python
class D2_MoE_Block(nn.Module):
    def __init__():
        self.Base_W       #Base Weight among all experts
        self.num_experts
        self.gate         #router function of MoE Block
        self.D2_experts = nn.ModuleList(D2_MoE_expert())
        # dynamic pruning function
        self.dynamic_pruner = two_phase_pruning(self.Base_W, X_calibration, target_sparsity)

    def forward(x):
        routing_weights = F.softmax(self.gate(x))

        # select expert mask for each token
        expert_mask = torch.top2(routing_weights, dim = 1)

        # dynamic pruning indice during inference
        pruned_indices = self.dynamic_pruner(x)
        # Base hidden states pre calculate for all experts with sparse inference
        Base_hidden_states = self.Base_W(x, pruned_indices)
        final_hidden_states = torch.zeros_like(x)

        for expert_idx in range(self.num_experts):
            current_expert = self.D2_experts[expert_idx]

            # selected token for current expert
            current_state = x[expert_mask]

            # correspond Base hidden states for current expert
            current_Base_hidden_states = Base_hidden_states[expert_mask]

            # calculate selected token hidden state with D2MoE expert
            current_hidden_states = current_expert(current_state,current_Base_hidden_states)

            # multiply current hidden state with router weights
            current_hidden_states *= routing_weights

            # add current hidden state to final hidden state
            final_hidden_states.add(current_hidden_states)
        return final_hidden_states

class D2_MoE_expert(nn.Module):
    def __init__():
        # low rank representation weights of Delta Weights
        self.Delta_U
        self.Delta_V

    def forward(x, current_Base_hidden_states):
        # get current_Base_hidden_states calculate from Base weights
        Delta_hidden_states = self.Delta_U(self.Delta_V(x))
        return current_Base_hidden_states + Delta_hidden_states
```

## C.3. Pytorch like Implement of two stage Semi-Dynamic Pruning Process

**Algorithm 3** Semi-Dynamic Pruning on Base Weight for D2MoE

```python
def two_phase_pruning(W_b, X, target_sparsity):
    """
    W_b: base weight matrix [m, n]
    X: input activations [n, b]
    target_sparsity: desired sparsity ratio (e.g., 0.5 means 50% pruned)
    """
    # Phase 1: Static Pruning
    static_metrics = []
    for j in range(n):
        # static pruning metric
        C_j = norm_L2(W_b[:, j]) * norm_L2(X[j, :])
        static_metrics.append((j, C_j))

    # sort the pruning metric
    static_metrics.sort(key=lambda x: x[1])
    static_k = int(n * target_sparsity * 0.5)
    static_pruned = set(idx for idx, _ in static_metrics[:static_k])

    # Phase 2: Dynamic Pruning
    def dynamic_prune(X_batch):
        dynamic_metrics = []
        for j in range(n):
            if j not in static_pruned:
                C_j = norm_L2(W_b[:, j]) * norm_L2(X_batch[j, :])
                dynamic_metrics.append((j, C_j))

        dynamic_metrics.sort(key=lambda x: x[1])
        dynamic_k = int(n * target_sparsity * 0.5)
        dynamic_pruned = set(idx for idx, _ in dynamic_metrics[:dynamic_k])

        return static_pruned | dynamic_pruned

    # return dynamic prune function
    return dynamic_prune
```

## C.4. Weighted Merge for Base Weights

**Algorithm 4** Delta Decomposition for MoE Models

0: **Input:** Expert weights $\{W_i\}_{i=1}^N$, Metric
0: **Output:** Base weight $W_b$, delta weights $\{\Delta W_i\}_{i=1}^N$
0: $W_b := \frac{\sum_{i \in N} \text{Metric}_i \cdot W_i}{\sum_{i \in N} \text{Metric}_i}$ {Frequency mean, Fisher-weighted mean, and other merging methods...}
0: **for** each expert $i \in \{1, \ldots, N\}$ **do**
0:     Compute $\Delta W_i = W_i - W_b$
0: **return** $W_b$, $\{\Delta W_i\}_{i=1}^N$ =0

## C.5. Truncation-Aware SVD Algorithm

**Algorithm 5** Truncation-Aware SVD Compression

0: **Input:** Delta weight $\Delta W_i$, activation matrix $X$
0: **Output:** Compressed delta weight $\Delta U_i, \Delta V_i$
0: Compute activation Gram matrix $S = XX^T$
0: Perform SVD: $\Delta W_i \cdot S = U \cdot \Sigma \cdot V^T$
0: Truncate $\Sigma$: $\Sigma^{\text{trunc}} = \text{TopK}(\Sigma)$
0: Reconstruct: $\Delta U_i = U \cdot \sqrt{\Sigma^{\text{trunc}}}$; $\Delta V_i = \sqrt{\Sigma^{\text{trunc}}} \cdot V^T \cdot S^{-1}$
0: **return** $\Delta U_i, \Delta V_i$ =0

