# OpenReview forum: "Delta Decompression  for MoE-based LLMs Compression"
_ICML.cc/2025/Conference — ICML 2025 poster_

### Official Review · Reviewer_4qJS · 2025-02-27

**Overall Recommendation:** 4

**Summary:**

The paper presents D²-MoE, a new compression framework designed to tackle issues of parameter redundancy, memory usage, and storage inefficiency in MoE LLMs. D²-MoE enhances efficiency by breaking down expert weights into a shared base weight, which captures common features, and a delta weight that represents expert-specific differences. It  employs truncation-aware Singular Value Decomposition to compress delta weights effectively and incorporates a semi-dynamic pruning strategy to remove redundancies while adapting to input distribution. Experimental results show that D²-MoE outperforms existing methods like NAEE and MoE-I², maintaining high accuracy and low perplexity even at high compression rates.

**Claims And Evidence:**

The claims made in paper are generally well-supported by experimental evidence, particularly those concerning the performance of the D²-MoE framework in achieving higher compression ratios and maintaining or improving accuracy compared to existing methods. The clear experiments demonstrating improvements in perplexity and other metrics provide credible evidence.

**Essential References Not Discussed:**

There might be relevant works on other sparsity of Transformers [1,2],  which are recommended for citation and discussion.

[1] Wang et al. Q-Sparse: All Large Language Models can be Fully Sparsely-Activated. CoRR abs/2407.10969 (2024).

[2] Li et al. The Lazy Neuron Phenomenon: On Emergence of Activation Sparsity in Transformers. ICLR 2023.

**Experimental Designs Or Analyses:**

The experiments in the paper are relatively well organized including different MoEs and different sparsities, as well as comparisons with different compression methods. Ablations and comprehensive and offer useful insights.

**Methods And Evaluation Criteria:**

The proposed methods for compressing MoE LLMs, including delta compression and semi-dynamical structured pruning, make sense for the problem at hand. They address well-known issues of redundancy and high memory usage in large-scale models. The evaluation criteria utilizing standard benchmark datasets (like WikiText-2, PTB, and C4) are robust, as they allow for meaningful performance comparisons against state-of-the-art methods.

**Other Comments Or Suggestions:**

This paper would benefit from deeper analysis of limitations.

**Other Strengths And Weaknesses:**

Pros:

1.	D²-MoE maintains that the delta parameter is novel and insightful in MoE merging.

2.	The performance results are compelling, demonstrating high efficiency and preservation of accuracy across different model scales.

3.	The writing and organization are relatively clear, making complex concepts understandable.


Cons:

1.	The theoretical framework lacks sufficient depth, particularly in justifying certain methodologies implemented within D²-MoE.

2.	The dynamic nature of reallocating delta weights might introduce additional complexity in implementation.

**Questions For Authors:**

1.	Authors could delve deeper into the mathematical principles behind truncation-aware SVD?

2.	Authors could discuss the potential for combining D^2-MoE with other advanced compression techniques, such as quantization or knowledge distillation.

**Relation To Broader Scientific Literature:**

The paper builds on existing literature about MoE architectures and LLM compression techniques.

**Theoretical Claims:**

The theoretical foundation requires further elaboration. Authors could delve deeper into the mathematical principles behind truncation-aware SVD.

---

> ### Author Rebuttal · Authors · 2025-03-30
>
> ### **Dear Reviewer 4qJS**
>
> Thank you for your insightful comments and for acknowledging the strengths of D²-MoE in terms of **efficiency, accuracy, and practical applicability**. Below, we address your concerns in detail.
>
> ------
>
> **Q1: Motivation and Theoretical Justification of the D²-MoE Framework**
>
> **A1:**
>
> As illustrated by the CKA similarity analysis in Figure 1, MoE models ***exhibit substantial similarities among different experts.*** Motivated by this observation, we propose merging experts into a shared representation. During this merging process, we utilize Fisher-weighted merging to emphasize critical expert weights effectively. Furthermore, ***to retain task-specific details and enhance model performance***, we introduce a delta branch by applying Singular Value Decomposition (SVD) to the delta weights.
>
> ------
>
> **Q2: The dynamic reallocation of delta weights might introduce additional implementation complexity—can this be analyzed further?**
>
> **A2:**
>
> (1) While delta weight reallocation **adds computational overhead**, Table 4 shows that **D²-MoE achieves up to 1.5× inference speedup** even with dynamic structured pruning.
>
> (2) Figure 2 demonstrates that **delta weights exhibit strong low-rank properties**, allowing for **efficient compression without excessive reallocation overhead**.
>
> (3) We plan to further refine caching or partial updates to reduce overhead, but for now the two-phase pruning strategy offers a favorable trade-off between complexity and improved compression.
>
> ------
>
> **Q3: The theoretical foundation of truncation-aware SVD should be elaborated on.**
>
> **A3:**
>
> (1) As shown in Table 6, ***naively truncating singular values leads to performance degradation and model collapse***. In contrast, truncation-aware SVD significantly outperforms both vanilla and activation-aware SVD, achieving a lower WikiText-2 perplexity (**5.28**) compared to standard SVD (**6.22**).
>
> (2) Therefore, we leverage the activation Gram matrix to compute a scale matrix represented as $\text{Cholesky}(X \cdot X^T)$. By introducing this scale matrix, truncating fewer singular values allows us to preserve more essential information.
>
> (3) Why we represent the scale matrix as $\text{Cholesky}(X \cdot X^T)$ can be proven as follows: When the scaling matrix $S$ is the Cholesky decomposition of $X \cdot X^T$, we have $S \cdot S^T = X \cdot X^T$. Under this condition, the compression loss $L_i$ caused by truncating singular values equals precisely to the singular value $sigma_i$ itself. Consequently, truncating the smallest singular values results in minimal compression loss, theoretically justifying our use of the scale matrix defined as $\text{Cholesky}(X \cdot X^T)$.
>
> ------
>
> **Q4: The discussion on combining D²-MoE with other compression techniques (e.g., quantization, knowledge distillation) should be expanded.**
>
> **A4:**
>
> (1) We integrate quantization to further reduce memory footprints in the following Table, following approaches like GPTQ that already show how delta weights can be quantized effectively. Additionally, we apply the mixed-precision quantization method from [**MC-MoE**](https://arxiv.org/abs/2410.06270) to our D²-MoE.
>
> ***Table D²-MoE+quantization***
>
> | Method                                 | WikiText-2↓ | PTB↓  | C4↓   | Openb. | ARC_e | WinoG. | HellaS. | ARC_c | PIQA  | MathQA | Average↑ |
> | :------------------------------------- | :---------- | :---- | :---- | :----- | :---- | :----- | :------ | :---- | :---- | :----- | :------- |
> | Mixtral-8x7B D²-MoE(25%) + GPTQ-4bit   | 5.34        | 22.03 | 9.56  | 0.288  | 0.761 | 0.741  | 0.556   | 0.460 | 0.771 | 0.345  | 0.56     |
> | Mixtral-8x7B GPTQ-3bit                 | 5.93        | 31.15 | 10.71 | 0.282  | 0.735 | 0.674  | 0.534   | 0.422 | 0.772 | 0.302  | 0.53     |
> | Mixtral-8x7B D²-MoE(40%) + MC-MoE-4bit | 5.42        | 22.71 | 9.85  | 0.286  | 0.742 | 0.730  | 0.541   | 0.423 | 0.766 | 0.331  | 0.55     |
>
>
>
> (2) Knowledge distillation can further enhance D²-MoE by **transferring knowledge from the full model to its compressed counterpart**, effectively preserving generalization capabilities. As demonstrated in the following table, applying advanced distillation methods indeed improves our approach's performance.
>
> ***Table D²-MoE+KD***
>
> | Method                        | WikiText-2↓ | PTB↓  | C4↓   | Openb. | ARC_e | WinoG. | HellaS. | ARC_c | PIQA  | MathQA | Average↑ |
> | :---------------------------- | :---------- | :---- | :---- | :----- | :---- | :----- | :------ | :---- | :---- | :----- | :------- |
> | Mixtral-8x7B D²-MoE(20%) + KD | 4.31        | 15.40 | 9.78  | 0.328  | 0.805 | 0.738  | 0.631   | 0.526 | 0.807 | 0.391  | 0.60     |
> | Mixtral-8x7B D²-MoE(40%) + KD | 4.69        | 21.61 | 10.74 | 0.318  | 0.792 | 0.726  | 0.603   | 0.506 | 0.795 | 0.354  | 0.58     |
> | Mixtral-8x7B D²-MoE(60%) + KD | 5.35        | 33.06 | 12.21 | 0.292  | 0.753 | 0.701  | 0.565   | 0.434 | 0.768 | 0.318  | 0.55     |

---

### Official Review · Reviewer_V5e9 · 2025-03-07

**Overall Recommendation:** 4

**Summary:**

This paper  decompose expert weights into a shared base weight and expert-specific delta weights, allowing for effective compression while preserving expert diversity. The delta weights are then compressed using SVD and the base weights undergo semi-dynamical structured pruning. The paper provides extensive empirical validation on Mixtral etc, demonstrating superior accuracy compared to prior methods.

## update after rebuttal

Thanks to the authors for their efforts to provide feedback. After the rebuttal, all of my concerns have been adequately addressed.

Thus, I tend to accept this submission.

**Claims And Evidence:**

Empirical results support the claim on performance, showing that D²-MoE consistently outperforms baselines.

**Essential References Not Discussed:**

None

**Experimental Designs Or Analyses:**

The experiments are comprehensive, covering various model sizes and compression ratios. The comparison across multiple baselines is a strong aspect of the paper, and the ablation on different merging methods (Table 5) is insightful.

**Methods And Evaluation Criteria:**

Evaluation covers multiple MoE models and diverse tasks. The use of WikiText-2, PTB, and C4 for language modeling perplexity evaluation, as well as ARC-e, PIQA, HellaSwag, and others tasks.

**Other Comments Or Suggestions:**

None

**Other Strengths And Weaknesses:**

**Strengths:**

+ The core idea of delta weight decomposition is novel and well-motivated.

+ Comprehensive experiments across multiple MoE models and datasets demonstrate clear performance gains.

+ High reproducibility with detailed pseudo-code, supplementary experiments, and claimed code availability.

**Weaknesses:**

+ Limited details on SVD truncation: how is the threshold determined dynamically?

+ Limited theoretical justification for Fisher merging: why is it better than other merge methods?

**Questions For Authors:**

Does the method generalize to non-MoE architectures? Could the same delta compression be used for compressing dense transformer models, or is it fundamentally MoE-specific?

**Relation To Broader Scientific Literature:**

The paper is well-aligned with recent advances in Mixture-of-Experts compression, building on methods like MoE-I², NAEE, and MoE-Compress.

**Theoretical Claims:**

The paper does not present significant new theoretical results but relies on well-established techniques like Fisher information weighting, SVD, and structured pruning. The Fisher-weighted merging is reasonable but would benefit from a more detailed theoretical justification of why it outperforms simpler merging strategies. The truncation-aware SVD is well-motivated, but a more formal discussion on how the truncation threshold is determined based on activation patterns would be useful.

---

> ### Author Rebuttal · Authors · 2025-03-30
>
> ### **Reviewer V5e9**
>
> Thank you for your detailed review and for recognizing the novelty of D²-MoE and its strong empirical performance. Below, we address your concerns in depth.
>
> ------
>
> **Q1: The criteria for setting the SVD truncation threshold should be explained.**
>
> **A1:**
>
> (1) We set an overall compression ratio (e.g., 40%), and to enhance performance, we prune 10% of the base weights. Under this setting, ***we can calculate the truncation threshold in D²-MoE accordingly.*** Detail can be seen in Table 7.
>
> (2) We integrate activation statistics by computing a scaling matrix from the activation Gram matrix (see Section3.3). This ensures that truncating fewer singular values preserves more essential information. And the scale matrix can be represented as $\text{Cholesky}(X \cdot X^T)$. And Table 6 demonstrates that truncation-aware SVD significantly outperforms vanilla and activation-aware SVD, reducing WikiText-2 perplexity to 5.28 compared to 6.22 for standard SVD.
>
> (3) Why we represent the scale matrix as $\text{Cholesky}(X \cdot X^T)$ can be proven as follows: When the scaling matrix $S$ is the Cholesky decomposition of $X \cdot X^T$, we have $S \cdot S^T = X \cdot X^T$. Under this condition, the compression loss $L_i$ caused by truncating singular values equals precisely to the singular value $sigma_i$ itself. Consequently, truncating the smallest singular values results in minimal compression loss, theoretically justifying our use of the scale matrix defined as $\text{Cholesky}(X \cdot X^T)$.
>
> ------
>
> **Q2: The theoretical justification for Fisher-weighted merging should be expanded—why is it better than simpler merging strategies?**
>
> **A2:**
>
> (1) We draw inspiration from model merging methods, where typically a base model is merged with various task-specific vectors. However, standard model merging techniques are ineffective for MoE expert merging since there is ***no explicit base expert weight.*** In contrast, ***Fisher merging directly operates on expert weights***, emphasizing parameters with higher gradient norms relative to the likelihood. This effectively identifies and retains the most critical expert weights.
>
>  (2) Our experiments in Table 5 confirm that  Fisher-weighted merging consistently outperforms mean averaging and frequency-based merging. Fisher merging achieves a 5.28 perplexity on WikiText-2, compared to 7.66 for mean averaging and 6.42 for frequency-based merging.
>
> ------
>
> **Q3: Does D²-MoE generalize to non-MoE architectures? Could this delta compression approach be applied to dense transformers?**
>
> **A3:**
>
> (1) We design D²-MoE specifically for multi-expert redundancy, but the principle of extracting a shared base plus compressed deltas is not limited to specialized experts.
>
> (2) We are now carrying out similar approach apply to large dense transformers ***by factoring out common subspaces from multiple layers and storing residual differences.***
>
> (3) We leave systematically exploring dense variants for future work, as the our D²-MoE  currently focuses on MoE layers.

---

### Official Review · Reviewer_LEQk · 2025-03-08

**Overall Recommendation:** 3

**Summary:**

This paper introduces D2-MoE, which decomposes expert weights into a shared base weight and unique delta weights. The delta weights are then compressed using SVD, and the base weight is further compressed using a semi-dynamical structured pruning strategy. The authors claim D2-MoE achieves better compression ratios and performance compared to other methods on models like Mixtral, Phi-3.5, DeepSeek, and Qwen2.

**Claims And Evidence:**

The claims are partially supported by the empirical results. The improvement against baseline methods is not very significant as Table 2 shows.

**Essential References Not Discussed:**

I suggest the authors carefully discuss the novelty of the proposed method against LoRA [1]. Additionally, the paper could discuss more recent work on the quantization method [2] for MoEs, as quantization is a common technique for further compressing LLMs.

[1] Hu, Edward J., et al. "Lora: Low-rank adaptation of large language models." ICLR, 2022.

[2] MC-MoE: Mixture Compressor for Mixture-of-Experts LLMs Gains More. ICLR 2025.

**Experimental Designs Or Analyses:**

I highly recommend the authors select some state-of-the-art baseline methods from top-tier machine learning conferences (i.e., ICML, NeurIPS, ICLR) for comparison, which makes their conclusion more convincing. Additionally, I suggest the authors use the conference version of each reference rather than the arxiv version.

**Methods And Evaluation Criteria:**

The proposed methods and evaluation criteria generally make sense.

**Other Comments Or Suggestions:**

N/A

**Other Strengths And Weaknesses:**

Overall, the proposed method is interesting and well motivates the MoE compression problem.

However, I have several concerns:

- The scalability of the proposed methods to even larger MoE models needs to be addressed. It would be great if the authors could do experiments on larger MoE models such as Deepseek V3.

- I would like to hear more about the novelty of this paper against LoRA which seems to leverage the delta weight idea as well.

- The selected baseline methods are relatively weak (from ACL or findings of ACL). I suggest the authors compare their method with some SOTA baselines from top-tier ML conferences.

- The paper should discuss more recent work on quantization methods for MoEs, as quantization is a common technique for further compressing large language models.

**Questions For Authors:**

See the weaknesses.

**Relation To Broader Scientific Literature:**

I am wondering how the proposed method relates to the idea of LoRA which also utilizes a delta weight.

**Theoretical Claims:**

N/A. There are no formal theoretical claims or proofs in the paper.

---

> ### Author Rebuttal · Authors · 2025-03-30
>
> ### **Dear Reviewer LEQk**
>
> **Q1: Careful discussion on relation with LoRA**
>
> **A1:**
>
> (1) Our framework structurally builds a multi-LoRA setup for MoE compression, consisting of a single base branch and multiple delta low-rank branches, enabling us to leverage existing LoRA research for further fine-tuning and ***efficient multi-LoRA inference***.
>
> (2) Unlike standard LoRA methods, our approach is a post-training compression framework that ***does not require additional training.***
>
> (3) Compared with similar methods such as [**LoSparse: Structured Compression of Large Language Models based on Low-Rank and Sparse Approximation(ICML 2023)**] and [**Merge, Then Compress: Demystify Efficient SMoE with Hints from Its Routing Policy (MC-sMoE, ICLR 2024 Spotlight)**], ***which also apply pruning and SVD but involve further training and both come from top-tier machine learning conferences***, our method significantly outperforms these approaches in performance and ***requires no training***, as illustrated in the table.
>
> ***Table D²-MoE vs LoSparse(ICML23), MC-sMoE(ICLR24 Spotlight)***
>
> | Method                     | WikiText-2↓ | PTB↓    | C4↓     | Openb. | ARC_e | WinoG. | HellaS. | ARC_c | PIQA  | MathQA | Average↑ |
> | :------------------------- | :---------- | :------ | :------ | :----- | :---- | :----- | :------ | :---- | :---- | :----- | :------- |
> | Mixtral-8x7B D²-MoE(20%)   | 4.65        | 16.32   | 8.59    | 0.33   | 0.80  | 0.75   | 0.61    | 0.51  | 0.81  | 0.39   | 0.60     |
> | Mixtral-8x7B MC-sMoE(20%)  | 5.00        | 15.36   | 9.68    | 0.336  | 0.794 | 0.766  | 0.603   | 0.503 | 0.794 | 0.380  | 0.59     |
> | Mixtral-8x7B MC-sMoE(40%)  | 4881.31     | 3276.94 | 4467.16 | 0.12   | 0.276 | 0.524  | 0.267   | 0.195 | 0.539 | 0.206  | 0.30     |
> | Mixtral-8x7B LoSparse(20%) | 953.51      | 805.16  | 1273.12 | 0.20   | 0.27  | 0.49   | 0.28    | 0.26  | 0.53  | 0.20   | 0.31     |
>
> (4) Additionally, further fine-tuning and knowledge distill to effectively recover performance, as shown in the table
>
> ***Table D²-MoE+further training***
>
> | Method                          | WikiText-2↓ | PTB↓  | C4↓   | Openb. | ARC_e | WinoG. | HellaS. | ARC_c | PIQA  | MathQA | Average↑ |
> | :------------------------------ | :---------- | :---- | :---- | :----- | :---- | :----- | :------ | :---- | :---- | :----- | :------- |
> | Mixtral-8x7B D²-MoE(40%) + KD   | 4.69        | 21.61 | 10.74 | 0.318  | 0.792 | 0.726  | 0.603   | 0.506 | 0.795 | 0.354  | 0.58     |
> | Mixtral-8x7B D²-MoE(40%) + LoRA | 4.54        | 16.04 | 9.17  | 0.31   | 0.771 | 0.739  | 0.604   | 0.463 | 0.789 | 0.348  | 0.57     |
>
> ------
>
> **Q2: Discussion on MoE quantization methods.**
>
> **A2:**
>
> (1) Quantization and D²-MoE are ***orthogonal approaches:*** Quantization aims to reduce the model's precision, primarily speeding up inference and reducing memory usage ***rather than parameter count***, while our D²-MoE targets parameter reduction explicitly through delta decomposition, it does acknowledge quantization as a complementary approach in the related work section, mentioning techniques like BitDelta that successfully quantize delta weights.
>
> (2) ***Existing quantization methods can be easily integrated into the D²-MoE framework.***
>
> ***Table D²-MoE+quantization***
>
> | Method                                 | WikiText-2↓ | PTB↓  | C4↓  | Openb. | ARC_e | WinoG. | HellaS. | ARC_c | PIQA  | MathQA | Average↑ |
> | :------------------------------------- | :---------- | :---- | :--- | :----- | :---- | :----- | :------ | :---- | :---- | :----- | :------- |
> | Mixtral-8x7B D²-MoE(25%) + GPTQ-4bit   | 5.34        | 22.03 | 9.56 | 0.288  | 0.761 | 0.741  | 0.556   | 0.460 | 0.771 | 0.345  | 0.56     |
> | Mixtral-8x7B D²-MoE(40%) + MC-MoE-4bit | 5.42        | 22.71 | 9.85 | 0.286  | 0.742 | 0.730  | 0.541   | 0.423 | 0.766 | 0.331  | 0.55     |
>
> ------
>
> **Q3: The scalability of D²-MoE to larger models**
>
> **A3:**
>
> (1) Table 2 show results on DeepSeekMoE-16B-Base at up to 60% compression. While not the exact DeepSeek V3 variant, these 16B-parameter experiments reveal consistent advantages, indicating D²-MoE scales to large MoE.
>
> (2) Due to limited experimental resources, we plan to integrate advanced DeepSeek V3 configurations in future work. To demonstrate the scalability of D²-MoE to larger models, ***we conduct experiments on Mixtral-8x22B.*** Experimental results are summarized in the table below:
>
> ***Table D²-MoE on larger scalability***
>
> | Method                    | WikiText-2↓ | PTB↓  | C4↓  | Openb. | ARC_e | WinoG. | HellaS. | ARC_c | PIQA | MathQA | Average↑ |
> | :------------------------ | :---------- | :---- | :--- | :----- | :---- | :----- | :------ | :---- | :--- | :----- | :------- |
> | Mixtral-8x22B             | 2.95        | 10.1  | 6.14 | 0.37   | 0.86  | 0.80   | 0.67    | 0.59  | 0.83 | 0.50   | 0.66     |
> | Mixtral-8x22B D²-MoE(20%) | 3.99        | 14.61 | 8.65 | 0.36   | 0.83  | 0.78   | 0.63    | 0.55  | 0.80 | 0.44   | 0.63     |

---

### Official Review · Reviewer_73q5 · 2025-03-10

**Overall Recommendation:** 4

**Summary:**

This paper introduces D²-MoE for MoE Language Models. The author decomposes expert weights into a shared base weight and expert-specific delta weights, then compresses each component separately.

**Claims And Evidence:**

The primary claim that D²-MoE outperforms existing compression methods is backed by comparative evaluations across multiple MoE architectures. The ablation studies further validate design choices.

**Essential References Not Discussed:**

A few additional references would strengthen the context:

"ZipLM: Inference-Aware Structured Pruning of Language Models" (Kurtic et al., 2023).

"Sheared LLaMA: Accelerating Language Model Pre-training via Structured Pruning" (Xia et al., 2023).

**Experimental Designs Or Analyses:**

The experimental design is comprehensive. I verified the methodology for evaluating MoE compression across different models, compression ratios, and benchmark tasks.

**Methods And Evaluation Criteria:**

The authors evaluate on both language modeling (perplexity on WikiText-2, PTB, and C4) and reasoning tasks (accuracy on seven reasoning benchmarks), providing a holistic assessment of model capabilities after compression.

**Other Comments Or Suggestions:**

The paper is well-written and organized, but could benefit from a few improvements:

1.	The introduction could more clearly separate the motivation (problems with existing approaches) from the proposed solution (D²-MoE components).\

2.	The figures showing CKA similarity and singular value energy retention could benefit from more detailed captions explaining the implications of these results.

3.	Minor typos: "experts delta decomposition" → "expert delta decomposition" (page 3), "Our D²-MoE successful compact" → "Our D²-MoE successfully compacts" (page 1).

**Other Strengths And Weaknesses:**

Strengths:

1.The delta decomposition approach is a new contribution to MoE compression. The method achieves substantial compression ratios.

2. The experiments on multiple models and benchmarks provide strong evidence for the method's effectiveness across diverse settings.

3.  D²-MoE doesn't require expensive fine-tuning after compression, making it more practical for large models.

Weaknesses:

1.Limited more systematic analysis of the relationship between compression ratio and performance degradation.

2.The paper doesn't examine how D²-MoE might interact with other compression techniques (like quantization) in a comprehensive compression pipeline.

**Questions For Authors:**

How does D²-MoE interact with quantization techniques? Since quantization is a common complementary approach to model compression, understanding whether these methods can be effectively combined would provide valuable insights for practitioners.

**Relation To Broader Scientific Literature:**

This work builds upon expertise from both MoE compression methods (like NAEE, MoE-I², and MoE-Pruner) and general LLM compression techniques.

**Theoretical Claims:**

The paper does not make formal theoretical claims requiring proofs, but rather presents empirically-grounded algorithmic innovations. The mathematical formulations are clearly presented.

---

> ### Author Rebuttal · Authors · 2025-03-30
>
> ### **Dear Reviewer 73q5**
>
> Thank you for your thoughtful review and constructive feedback. We appreciate your recognition of the novelty of D²-MoE and its strong empirical performance. Below, we address your concerns in detail.
>
> **Q1: The relationship between compression ratio and performance degradation is not analyzed in detail.**
>
> **A1:**
>
> (1) Table 2 demonstrates D²-MoE's robustness at various compression ratios. For example, on Mixtral-8×7B, performance declines gradually with increased compression (from 20% to 60%) without model collapse. Specifically, at **40% compression**, D²-MoE achieves an average accuracy of **0.57**, significantly outperforming NAEE (**0.48**) and MoE-I² (**0.49**). Even at **60% compression**, D²-MoE maintains **0.52** accuracy, whereas NAEE drops sharply to **0.36**.
>
> (2) Figure 4 shows how delta weight trimming affects D²-MoE's performance. As trimming increases, perplexity gradually rises, yet our method maintains stable performance even at extreme compression ratios of **81%**. Specifically, at **43%** compression (trimming 1 delta weight), D²-MoE achieves a WikiText-2 perplexity of **6.43**. Remarkably, even under extreme 81% compression, the model does not collapse, demonstrating the robustness and effective trade-off between compression ratio and performance.
>
> ------
>
> **Q2: The interaction of D²-MoE with other compression techniques, particularly quantization, is not explored.**
>
> **A2:**
>
> (1)  ***Our current approach combines SVD decomposition and pruning, orthogonal to quantization.*** We focus on decomposing experts into Fisher-weighted bases and SVD-compressed delta weights without formally integrating quantization yet. However, our design allows straightforward quantization of low-rank delta factors ***(Section “Delta compression in MoE LLMs”).***
>
> (2) ***Existing quantization methods can be easily integrated into the D²-MoE framework:*** We plan to integrate quantization techniques to further reduce memory footprint, as demonstrated in the following table. Approaches such as GPTQ have shown effective quantization of delta and base-merged weights. Additionally, we apply the mixed-precision quantization method from [**MC-MoE**](https://arxiv.org/abs/2410.06270) to our D²-MoE.
>
> ***Table D²-MoE+quantization***
>
> | Method                                       | WikiText-2↓ | PTB↓  | C4↓   | Openb. | ARC_e | WinoG. | HellaS. | ARC_c | PIQA  | MathQA | Average↑ |
> | :------------------------------------------- | :---------- | :---- | :---- | :----- | ----- | :----- | :------ | :---- | :---- | :----- | :------- |
> | Mixtral-8x7B D²-MoE(25%) + GPTQ-4bit         | 5.34        | 22.03 | 9.56  | 0.288  | 0.761 | 0.741  | 0.556   | 0.460 | 0.771 | 0.345  | 0.56     |
> | Mixtral-8x7B GPTQ-3bit                       | 5.93        | 31.15 | 10.71 | 0.282  | 0.735 | 0.674  | 0.534   | 0.422 | 0.772 | 0.302  | 0.53     |
> | Mixtral-8x7B D²-MoE(40%) + MC-MoE-4bit       | 5.42        | 22.71 | 9.85  | 0.286  | 0.742 | 0.730  | 0.541   | 0.423 | 0.766 | 0.331  | 0.54     |
> | DeepSeekMoE-16B-Base D²-MoE(25%) + GPTQ-4bit | 7.62        | 12.61 | 12.94 | 0.264  | 0.707 | 0.655  | 0.511   | 0.373 | 0.769 | 0.275  | 0.51     |
> | DeepSeekMoE-16B-Base GPTQ-3bit               | 8.33        | 13.79 | 16.01 | 0.252  | 0.677 | 0.653  | 0.445   | 0.358 | 0.711 | 0.269  | 0.48     |
>
>
>
> ------
>
> **Q3: The introduction should better separate the motivation from the proposed solution.**
>
> **A3:**
>
> (1) ***Problems with existing approaches:*** We will revise the introduction to juxtapose the storage/memory challenges of MoE (motivation) with our delta decomposition approach (solution), referencing new Table 1 (Section “Related Work”) to highlight how D²-MoE differs from pure pruning or merging.
>
> (2) ***D²-MoE's motivation:*** We emphasize that the moderate overlap (CKA 0.3–0.5) between experts to extract a shared base weight while preserving expert-specific diversity in compressed delta form.
>
> ------
>
> **Q4: Figures (e.g., CKA similarity, singular value energy retention) need more detailed captions explaining their implications.**
>
> **A4:**
>
> (1) We will **expand the captions** of Figures 3 and 4 to explicitly describe what each metric represents.
>
> (2) For **Figure 3 (CKA similarity)**, we will clarify that **lower similarity values indicate that merging all experts directly would lead to performance loss**, justifying the need for delta decomposition.
>
> (3) For **Figure 4 (singular value energy retention)**, we will highlight that **delta weights exhibit strong low-rank properties**, making them well-suited for **SVD-based compression**.
>
> ------
>
> **Q5: Minor typos need correction.**
>
> **A5:**
>
> (1) We will correct all typos, including those in **page 1 ("successful compact" → "successfully compacts")** and **page 3 ("experts delta decomposition" → "expert delta decomposition")**.
>
> (2) We will conduct a **thorough proofreading pass** to ensure consistency and clarity throughout the paper.

---

### Decision · Program_Chairs · 2025-05-01

**Decision:**

Accept (poster)

**Comment:**

This paper introduces a new method to decompose expert weights in MoE models into a shared base weight and expert-specific delta weights using SVD. By doing this, the paper has shown that it could achieve higher quality than other MoE layer compression methods.

All four reviewers had a consensus that the proposed method was able to effectively compress MoE layers. Most of the reviewers also appreciated its novelty and comprehensive evaluations. As a result, all four reviewers agreed on accepting the paper (3 accepts and 1 weak accept).

There were a few clarification questions, but the authors provided answers to resolve those.

Overall, this paper will be a good contribution to the conference and community.